# Optimal Design of Piezoelectric Cantilevered Actuators for Charge-Based Self-Sensing Applications

**DOI:** 10.3390/s19112582

**Published:** 2019-06-06

**Authors:** Joël Bafumba Liseli, Joël Agnus, Philippe Lutz, Micky Rakotondrabe

**Affiliations:** Department of Automatic Control and Micro-Mechatronic Systems, FEMTO-ST Institute, Université de Bourgogne Franche-Comté, CNRS, 24 rue Savary, F-25000 Besançon, France; joel.agnus@femto-st.fr (J.A.); philippe.lutz@femto-st.fr (P.L.); mrakoton@femto-st.fr (M.R.)

**Keywords:** design, optimization, piezoelectric actuators and sensors, self-sensing actuation, micro-/nano-robots

## Abstract

Charge-based Self-Sensing Actuation (SSA) is a cost and space-saving method for accurate piezoelectric based-actuator positioning. However, the performance of its implementation resides in the choice of its geometry and the properties of the constituent materials. This paper intends to analyze the charge-based SSA’s performances dependence on the aforementioned parameters and properties for a piezoelectric cantilever. A model is established for this type of Piezoelectric Actuator (PEA), and a multi-objective function is defined. The multi-objective function consists of the weighted actuator and sensor objective functions of the PEA. The analytical optimization approach introduced herein aims to assess the evolution of the defined multi-objective function across a defined set of geometrical parameters and material properties and highlights the existence of a subset of solutions for an optimal charge-based SSA’s implementation. The commercially-available finite element analysis software, COMSOL Multiphysics, is used on the parametric model of the given structure to validate the analytical model. Then, experiments are conducted to corroborate the numerical and analytical modeling and analysis.

## 1. Introduction

The sensor requirements for nanopositioning systems are among the most demanding of any control system. The sensors must be compact, high-speed, and able to resolve position down to the nanoscale [1]. By allowing a Piezoelectric Actuator (PEA) to be simultaneously an actuator while measuring its displacement (strain) and/or the perceived force (stress) as its own sensor, the implementation of the Self-Sensing Actuation (SSA) is an inexpensive and space-saving solution for an accurate nanopositioning system. SSA provides a sensor that meets all of the above requirements at no cost or extra space occupancy. In the past few decades, SSA has gained attention in micro-engineering applications by enabling the design of compact devices with lower costs and simpler configurations [2,3,4].

Overall, there are two SSA approaches:*SSA based on the piezoelectric direct effect*, which consists of a dual dielectric feedthrough cancellation to recover the signal arising from the PEA’s strain. It is achieved either by using a capacitance bridge where one represents the PEA as a strain-dependent voltage source (voltage-based SSA) [5] or with an antiparallel circuit when one chooses to represent the PEA as a strain-dependent charge source (charge-based SSA) [6,7].*SSA based on the PEA’s change of electrical properties*, that is by measuring the changing relative permittivity [8], capacitance [9], or capacitance and resistance at the same time [10] to estimate the PEA’s displacement and/or perceived force.

For this paper, only the charge-based SSA will be considered, because of its simple underlying principle and easy implementation [3,11,12,13].

Even though charge-based SSA has demonstrated a great potential [3,11,12,13], its effectiveness depends on the morphology of the PEA and the choice of the mechanical properties of its constituent materials. Therefore, the first step to take full advantage of the potential of charge-based SSA is the PEA’s design. The design of the piezoelectric bender (see Figure 1) should maximize both the bender’s tip displacement $\delta_{pre}$ (or strain in the x−direction
$S_{1_{pre}}$) due to a control voltage $v_{c}$ and the amount of strain-induced charges $Q_{\delta }$ due to external forces $F_{ext}$ and/or piezoelectric bending (bending due to the piezoelectric reverse effect under the application of $v_{c}$).

### 1.1. Piezoelectric Actuator Design Optimization

The distribution of material inside the piezoelectric layers influences the actuators’ performance, and hence, the number, shape, size, and placement of the actuators have to be optimized. Among the optimization methods for piezoelectric actuators, there are:Parametric optimization in which the parameters of the PEA are varied in order to determine the dimensions and material properties that guarantee improved performances in terms of output range [14,15] $\left(\frac{\delta_{pre}}{v_{c}}\right)$ and in terms of bandwidth [16]. $\delta_{pre}$ is the bending due to the piezoelectric reverse effect under the application of $v_{c}$. Another parametric optimization for PEAs is found in [17] where the design of the dimensions of the PEA that would satisfy prescribed performances are based on interval techniques [18].Topology optimization, which is based on the Piezoelectric Material with Penalization (PEMAP) model, where the design variable is the pseudo-density $\rho_{1}$, which describes the amount of piezoelectric material in each finite element in the piezoelectric layer(s). Topology optimization is employed to find an optimal distribution of piezoelectric material in a multi-layer plate or shell structure to provide the maximum displacement $\delta_{pre}$ or generated forces in a given direction at a given point of the domain [19,20,21].Simultaneous topology and polarization optimization, which uses the Piezoelectric Material with Penalization and Polarization (PEMAP-P) and works to find the optimum actuator layout and polarization profile simultaneously [22,23,24]. For this method, in addition to the pseudo-density $\rho_{1}$, a new design variable $\rho_{2}$ is introduced for the polarization of the piezoelectric material. The optimization problem consists of distributing the piezoelectric actuators in such a way so as to achieve a maximum output displacement $\delta_{pre}$ in a given direction at a given point of the structure, while simultaneously minimizing the structural compliance.

All these methods aim to determine the geometric feature that will enhance the PEA’s generated displacement due to an applied input voltage without regard to the sensor aspect of the piezoelectric material.

### 1.2. Piezoelectric Sensor Optimization

On the one hand, applying an input voltage will result in the piezoelectric material elongation/contraction, and on the other hand, a pressure applied onto a piezoelectric material will be converted into an electrical output (strain-induced charges). When a piezoelectric material is used to convert mechanical into electrical energy, it is called a piezoelectric sensor. As for piezoelectric actuators, works have been conducted for the optimization of piezoelectric sensors.

These searches can be grouped into two categories:Geometric optimization methods that aim at the optimization of geometric parameters such as the length, width, and thickness of the piezoelectric layer in order to maximize the output recuperated electrical charges at the electrodes $Q_{\delta }$. Pillai et al. [6] presented an analytical method to design an optimal unimorph beam that maximizes the strain-induced charges’ sensitivity when acted upon by a uniform mechanical load *p* at a specified frequency $\nu_{s}$
$\left(\frac{Q_{\delta }}{p(\nu_{s})}\right)$. Schlinquer et al. [25] suggested a unimorph, and Chen et al. [26] a bimorph piezoelectric cantilever mechanical structure optimization for energy harvesting. The optimized design aimed to maximize the PEA’s strain-induced charges due to external harmonic load $\left(\frac{Q_{\delta }}{F_{ext}\left(\nu \right)}\right)$, where ν is the frequency of the harmonic load.Localization optimization methods that aim to find the placement with the highest pressure point on a given structure and thus guarantee the highest possible output recuperated electrical charges at the electrodes [27,28].

### 1.3. Simultaneous Piezoelectric Actuation and Sensing Design Optimization

Some researchers have considered using both the direct and reverse effect of piezoelectric materials to ensure simultaneous good observability and controllability of a structure. Moheimani et al. [29] proposed an electrode pattern on a piezoelectric tube actuator for simultaneous sensing and actuation. Moussa et al. [30] used a topological optimization method to design a compliant microactuator that optimally integrates actuating and sensing areas in a monolithic structure. Rougeot et al. [31] introduced a three-layered piezoelectric cantilever design for which the upper and lower layers were used for the PEA’s actuation, whilst the middle layer served for the sensing of the PEA’s displacement and perceived force. In the approaches of [29,30,31], the actuation and sensing did not share the same electrodes. Therefore, the resulting structure cannot be considered to be an optimized PEA design for charge-based SSA.

Masson et al. [32] presented an analytical approach for the design of piezoelectric cantilever actuators, which aims to improve the SSA performance for external loads’ estimation while minimizing the dielectric effect $\left(C_{pea}\cdot v_{c}\right)$. The optimized piezoelectric cantilever mechanical structure they proposed was supposed to achieve a tip displacement of at least $\delta^{min}=25\;\mu m$, a minimum blocking force of $F_{bl}^{min}=100$ mN, and withstand a maximum electric field of $E_{3}^{max}=3$ V/μm (depolarizing field). The optimization problem was formulated as follows:
(1)$$\begin{array}{c} \underset{l,w,h}{\mathrm{minimize}}\;\;F\left(l,w,h\right)=-\frac{Q_{\delta }}{F_{ext}\left(C_{pea}\cdot v_{c}\right)}\\ \mathrm{subject}\;\mathrm{to}\;\;\;E_{3}\left(h_{p}\right)⩽3\;V/\mu m\\ \;\;\;\;\;\;\;\;\;F_{bl}\left(l,w,h\right)⩾100\;\mathrm{mN}\\ \;\;\;\;\;\;\;\delta (l,w,h)⩾25\;\mu m \end{array}$$
where $C_{pea}$ is the PEA’s capacitance in the absence of mechanical deformation and null electric field.

The approach suggested an optimization for the external load estimation $\left(\frac{Q_{\delta }}{F_{ext}}\equiv \frac{Q_{\delta }}{\delta_{mec}}\right)$ and did not account for the piezoelectric actuation-induced charges $\left(\frac{Q_{\delta }}{\delta_{pre}}\right)$. A simplified schematic of the superposition of bending under external load and piezoelectric bending is shown in Figure 2 to help perceive the difference between both. Furthermore, the objective function F(l,w,h) just constrain the minimum displacement and blocking force and does not aim to maximize the actuation $\left(\frac{\delta_{pre}}{v_{c}}\right)$.

Earlier studies on the optimal design of piezoelectric structures aimed to optimize the sensor’s sensitivity to mechanical loads and to increase the actuation. None intended to optimize both simultaneously. In the following is presented an approach for the optimal design of the morphology and choice of the mechanical properties of a unimorph piezoelectric cantilever for optimized sensor and actuator properties simultaneously.

The remainder of the article is organized as follows. In Section 2, the principle of SSA is briefly discussed and the constitutive equations of a unimorph piezoelectric cantilever for piezoelectric actuation and strain sensing are presented. Then, in Section 3, we introduce the parameters’ optimization function for actuation, sensing, and a multi-objective function that allows optimal actuation and strain sensing at the same time. In Section 4 and Section 5, respectively, finite element analysis and experiments results of the given structure are presented, as well as a comparison with the results of the analytical model. Finally, conclusions are provided in Section 6.

## 2. Self-Sensing Actuation Analytical Model

The SSA technique allows a single piece of piezoelectric material to concurrently sense and actuate in a closed loop system. A key characteristic of these materials is the use of the piezoelectric reverse effect to actuate the structure in addition to the direct effect to sense structural deformation. Indeed, mechanical stress provokes the appearance of electrical charges on the material’s surface (piezoelectric direct effect), and an electric field provokes the deformation of the material (piezoelectric reverse effect). Due to the reverse and direct effects, the electrodes used to supply the PEA can also be used to retrieve the charges appearing due to the deformation thereof. In an SSA’s implementation, the drive and sense electrodes share a common node. One benefit of a self-sensing actuator is that the sensor and actuator are truly collocated. Collocated control has been shown to have a number of advantages relating to the closed-loop stability of the structure [33].

Before attempting to optimize the design of a piezoelectric material for SSA’s implementation, it is necessary to establish its constitutive relationships. These constitutive relationships to describe a piezoelectric material’s electromechanical behavior are compiled in two main equations:The charge equation, which models the electrical subsystem of the PEA. This equation describes how the charge flowing through the PEA is split into a dielectric charge $Q_{d}$ and a charge $Q_{\delta }$ that is related to the mechanical deformation of the PEA.The displacement equation, which models the mechanical subsystem of the PEA. This equation describes the contribution of external forces $F_{ext}$ and applied voltage $v_{c}$ to the generated displacement by the PEA δ.

An optimal piezoelectric sensor would maximize the charges resulting from the PEA’s deformation $\left(\frac{Q_{\delta }}{\delta }\right)$. An optimal piezoelectric actuator would maximize the displacement generated by the applied voltage $\left(\frac{\delta }{v_{c}}\right)$. However, piezoelectric actuators or sensors are electromechanically-coupled systems. Any change of the electrical parameters of a piezoelectric element reverberates on the mechanical side and vice versa. This intermingling of electric and elastic phenomena will determine the performances of the SSA’s implementation for which both $Q_{\delta }$ and δ must be optimized concurrently.

In the following subsections, the charge and displacement equations are going to be determined for the case of a unimorph piezoelectric cantilever.

### 2.1. Charge Equation (Strain Sensitivity)

Consider the unimorph in Figure 1. To improve the SSA performance for the estimation of external loads and deformations due to piezoelectric bending, it is necessary to optimize the position of $h_{0}$ and $h_{\gamma }$, the neutral plane (The neutral plane is the surface within the beam, where the material of the beam is not under stress, neither compression nor tension) under external load, and piezoelectric bending, respectively. The derivation of both terms for a unimorph piezoelectric cantilever can be found in [34,35] and are given by:(2)$$\begin{array}{c} h_{0}=\frac{h_{e}^{2}s_{11}^{p}+h_{p}^{2}s_{11}^{e}+2h_{e}h_{p}s_{11}^{e}}{2(h_{e}s_{11}^{p}+h_{p}s_{11}^{e})} \end{array}$$(3)$$\begin{array}{c} h_{\gamma }=-\frac{3h_{p}h_{e}^{2}s_{11}^{p}+2h_{e}^{3}s_{11}^{p}-h_{p}^{3}s_{11}^{e}}{6h_{e}s_{11}^{e}\left(h_{e}+h_{p}\right)} \end{array}$$
where *s* is the material compliance modulus, *h* the layer thickness, and the subscripts (or superscripts) *p* and *e* denote the piezoelectric and elastic layer, respectively.

From the optimized position of $h_{0}$ and $h_{\gamma }$ will result a mostly positive (or negative) stress distribution in the PEA’s cross-sections (see Figure 3). The stress distribution in the PEA’s cross-sections is directly linked to the strain-induced charge $Q_{\delta }$. Indeed, piezoelectric materials are considered to be a stack of infinitesimal small layers, and each one contributes to the overall charge available at the PEA’s electrodes $Q_{\delta }$ according to the stress in the layer $T_{1}\left(x,y,z_{p}\right)$ [34,35]. A mostly positive (or negative) stress distribution will result in a better ratio between δ, the bender’s tip displacement, and $Q_{\delta }$. From a good ratio $\frac{Q_{\delta }}{\delta }$ will result a good estimation sensitivity to both external loads and strain due to the piezoelectric actuation.

The analytical model to be used herein was extracted from [34,35,36]. In [34,35,36], the Poisson effect was neglected, that is only the stress in the x−direction, $T_{1}\left(x,y\right)$, was considered and $T_{i=2,\cdots ,6}\left(x,y\right)$ neglected. If one applies an electric field $E_{3}=\frac{v_{c}}{h_{p}}$ and $F_{ext}$ at the tip of the PEA, both in the poling direction, the total amount of charges accumulated on the PEA’s electrodes is given by: (4)$$Q_{pea}=Q_{\delta }+Q_{d}={∯}_{A_{E}}\left(\underset{D_{3}}{\underbrace{d_{31}T_{1}\left(x,y\right)+\varepsilon_{33}^{T}E_{3}}}\right)\mathrm{dA}$$
where $A_{E}=l\cdot w$ is the PEA’s electrode area and *l* and *w* are the length and width thereof (see Figure 1). It is worth mentioning that the PEA’s electrodes surface is always equal to the PEA’s surface. $Q_{d}$ is compensated by the charge-based SSA algorithm with the knowledge of $v_{c}$ [7,13], so that the charge to be used for the estimation is only related to the external load and the strain due to the piezoelectric actuation.

The PEA’s effective stress ($T_{1}\left(x,y\right)=T_{1}\left(x\right)$
∀y) in Equation (4) results from piezoelectric actuation $T_{1_{pre}}$ and bending due to external loads $T_{1_{mec}}$. The piezoelectric bending $\delta_{pre}$ results from the compression or extension in the piezoelectric material enforced by the electric field $E_{3}$. This is the mechanical base of all piezoelectric actuation. For more details, see [34,35]. For the bending due to external loads $\delta_{mec}$, the piezoelectric bender behaves as a passive composite beam [37].

From Equation (4), one can deduce the total amount of strain-induced charges due to both the external load and the piezoelectric bending as:(5)$$\begin{array}{c} Q_{\delta }\left(l,w,h_{e},h_{p},s_{11}^{e},s_{11}^{p}\right)=d_{31}\int_{0}^{l}\int_{0}^{w}\left[T_{1_{pre}}\left(h_{e},h_{p},s_{11}^{e},s_{11}^{p}\right)+\right.\\ \left.T_{1_{mec}}\left(l,w,h_{e},h_{p},s_{11}^{e},s_{11}^{p}\right)\right]\mathrm{dxdy} \end{array}$$

From the calculations by Smits et al. [36], the effective stress of the piezoelectric layer of a piezoelectric bender’s cross-section is given by:(6)$$\begin{array}{c} T_{1_{pre}}\left(h_{e},h_{p},s_{11}^{e},s_{11}^{p}\right)+T_{1_{mec}}\left(l,w,h_{e},h_{p},s_{11}^{e},s_{11}^{p}\right)=\\ \frac{1}{h_{p}}\int_{h_{p_{-}}}^{h_{p_{+}}}\left\{s_{11}^{e}{\left(h_{p}\right)}^{3}+s_{11}^{p}{\left(h_{e}\right)}^{3}-6s_{11}^{e}h_{p}\left(h_{e}+h_{p}\right)\left[z_{p}-\frac{h_{p}}{2}\right]\right\}\frac{d_{31}h_{e}}{K\left(h_{e},h_{p},s_{11}^{e},s_{11}^{p}\right)h_{p}}v_{c}{\mathrm{dz}}_{p}\\ -\frac{1}{h_{p}}\int_{h_{p_{-}}}^{h_{p_{+}}}\left[2\left(s_{11}^{p}h_{e}+s_{11}^{e}h_{p}\right)z_{p}+s_{11}^{p}{\left(h_{e}\right)}^{2}-s_{11}^{e}{\left(h_{p}\right)}^{2}\right]\frac{6s_{11}^{e}}{K(h_{e},h_{p},s_{11}^{e},s_{11}^{p})w}M\left(x\right){\mathrm{dz}}_{p} \end{array}$$
where $K(h_{e},h_{p},s_{11}^{e},s_{11}^{p})$ is given by:(7)$$\begin{array}{c} K\left(h_{e},h_{p},s_{11}^{e},s_{11}^{p}\right)=4s_{11}^{p}s_{11}^{e}h_{e}{\left(h_{p}\right)}^{3}+4s_{11}^{p}s_{11}^{e}{\left(h_{e}\right)}^{3}h_{p}+{\left(s_{11}^{p}\right)}^{2}{\left(h_{e}\right)}^{4}+{\left(s_{11}^{e}\right)}^{2}{\left(h_{p}\right)}^{4}\\ +6s_{11}^{p}s_{11}^{e}{\left(h_{e}\right)}^{2}{\left(h_{p}\right)}^{2} \end{array}$$
and M(x) is the bending moment in which all occurring external loads will be merged. For this study, one only considers an external force $F_{ext}$ applied at the tip of the bender, that is $M\left(x\right)=F_{ext}\cdot \left(l-x\right)$.

To optimize the SSA’s sensitivity to strain due to the piezoelectric converse effect, one needs to optimize $\left(\frac{Q_{\delta }}{v_{c}}\equiv \frac{Q_{\delta }}{\delta_{pre}}\right)$. To optimize the SSA sensitivity to strain due to mechanical load, one needs to optimize $\left(\frac{Q_{\delta }}{F_{ext}}\equiv \frac{Q_{\delta }}{\delta_{mec}}\right)$.

### 2.2. Displacement Equation (Actuation)

The displacement of the tip of the piezoelectric cantilever can be found by applying the principle of superposition: the deformation contributed by the mechanical flexure of the beam due to the application of an external force $F_{ext}$ is added to the deformation due to the piezoelectric converse effect under the application of $v_{c}$. The sum of the two displacements is equal to the total displacement of the tip of the bender (see Figure 2).

Calculations to derive the displacement equation of the piezoelectric cantilever can be found in [36,38]. The displacement due to the superposed effect of $v_{c}$ and $F_{ext}$ is given by:(8)$$\delta \left(l,w,h_{e},h_{p},s_{11}^{e},s_{11}^{p}\right)=\left[\frac{4(s_{11}^{p}h_{e}+s_{11}^{e}h_{p})l}{w}F_{ext}-3d_{31}h_{e}\left(h_{e}+h_{p}\right)v_{c}\right]\frac{s_{11}^{e}s_{11}^{p}l^{2}}{K(h_{e},h_{p},s_{11}^{e},s_{11}^{p})}$$
One also knows from [36,38] that:(9)$$\frac{\delta }{v_{c}}\left(l,h_{e},h_{p},s_{11}^{e},s_{11}^{p}\right)=\frac{Q_{\delta }}{F_{ext}}\left(l,h_{e},h_{p},s_{11}^{e},s_{11}^{p}\right)$$
Besides the tip deflection δ, the tip force that can be generated by the PEA, also referred to as blocking force, is of interest. From Equation (8) can be derived the PEA’s blocking force $F_{bl}$ as follows:(10)$$F_{bl}\left(l,w,h_{e},h_{p},s_{11}^{e},s_{11}^{p}\right)=\frac{3}{4}\frac{d_{31}h_{e}\left(h_{e}+h_{p}\right)w}{(s_{11}^{p}h_{e}+s_{11}^{e}h_{p})l}v_{c}$$

$F_{bl}$ is used to define the maximum force generated by the actuator. As suggested by [32], $F_{bl}$ can also be used to impose a geometric constraint and help reduce the number of parameters to be fine-tuned. Using commercial actuator benders from PIceramic as a reference, we opted for $F_{bl}=100\;mN$ with an applied input voltage of $v_{c}=60\;V$. This yields, from Equation (10):(11)$$w\left(l,h_{e},h_{p},s_{11}^{e},s_{11}^{p}\right)⩾0.0022\frac{(s_{11}^{p}h_{e}+s_{11}^{e}h_{p})l}{d_{31}h_{e}\left(h_{e}+h_{p}\right)}$$

Equation (11) expresses the geometric constraint that guarantees the minimal blocking force $F_{bl}$.

## 3. Optimization of the Design

### 3.1. Definition of the Multi-Objective Function

Regardless of whether it is a piezoelectric stack, piezoelectric cantilever, or piezoelectric tube actuator, the PEA’s physical parameters to be fine-tuned will alter the stiffness of the actuator to allow the maximum displacement possible while ensuring a better internal stress distribution within the piezoelectric layers for strain-induced charged measurement. Apart from piezoelectric stack actuators, for which the entire piezoelectric material undergoes tensile or compressive stress, in bending piezoelectric based-actuators such as the piezoelectric cantilever and piezo tube actuators, the internal stress distribution of the PEA’s cross-section may contain both tensile and compressive stress for a given cross-section of the PEA (see Figure 3). The internal stress distribution of the PEA’s cross-section is of the utmost importance for strain-induced charge measurement. The total accumulated charges on the PEA’s electrodes is the contribution of all the Weiss domains constituting the piezoelectric material. The generated charge (positive or negative) of each Weiss domain is a function of the stress (tensile or compressive) undergone by it [34,35].

When designing a PEA for charge-based SSA’s implementation, one should optimize the PEA’s morphology to not only maximize the displacement due to the piezoelectric actuation $\left(f_{1}=\frac{\delta }{v_{c}}{|}_{F_{bl}⩾100\;mN}\right)$, but also the charges produced by the PEA’s strain. The PEA’s strain may result from piezoelectric actuation $\left(f_{2}=\frac{Q_{\delta }}{v_{c}}{{|}_{F_{bl}⩾100\;mN}\equiv \frac{Q_{\delta }}{\delta_{pre}}|}_{F_{bl}⩾100\;mN}\right)$, external loads $\left(f_{3}=\frac{Q_{\delta }}{F_{ext}}\equiv \frac{Q_{\delta }}{\delta_{mec}}\right)$, or the superposition of both effects. The optimized design of a PEA’s morphology for SSA should result from the optimization of the scalarized (Scalarizing a multi-objective optimization problem is an a priori method, which means formulating a single-objective optimization problem such that optimal solutions to the single-objective optimization problem are Pareto optimal solutions to the multi-objective optimization problem) set of the actuator and sensor objectives into a single objective. Furthermore, from Equation (9), one knows that an optimal $f_{1}$ is equivalent to an optimal $f_{3}$. Consequently, the multi-objective function to be maximized can be written as follows:(12)$$\begin{array}{c} \underset{l,w,h_{e},h_{p},s_{11}^{e},s_{11}^{p}}{\mathrm{minimize}}\;\;F\left(l,h_{e},h_{p},s_{11}^{e},s_{11}^{p}\right)=-\sum_{i=1}^{2}W_{i}{\hat{f}}_{i}\\ \mathrm{subject}\;\mathrm{to}\;w(l,h_{e},h_{p},s_{11}^{e},s_{11}^{p})⩾0.0022\frac{(s_{11}^{p}h_{e}+s_{11}^{e}h_{p})l}{d_{31}h_{e}\left(h_{e}+h_{p}\right)}\\ \;\;\;\;\;\;\;\;E_{3}\left(h_{p}\right)⩽E_{max} \end{array}$$
where $W_{i}{|}_{i=1,2}$ are user-supplied weights and ${\hat{f}}_{i}=\frac{f_{i}}{|f_{i}|}$ is the normalized vector of $f_{i}$.

### 3.2. Analysis of the Objective Function

The piezoelectric material to be used for this optimization is the PSI-5H4E (Initial depolarizing field $E_{c}∼0.3$ V/μm, $d_{31}=-320\times 10^{-12}\;$ C/N, $d_{33}=650\times 10^{-12}$ C/N, $s_{11}^{p}=1.6129\times 10^{-11}\;$m${}^{2}$/N). Equation (8) states that $\delta \;\propto \;v_{c}$, that is the higher the applied voltage, the higher the bender tip’s displacement. Moreover, from Equation (11), one knows that $v_{c_{max}}=60$ V. Consequently, using $E_{c}$ as a limiting factor, one can determine the piezoelectric layer thickness, that is:(13)$$h_{p}=\eta \frac{v_{c_{max}}}{E_{c}}$$
where η is the safety factor. For η=0.5
$\Rightarrow h_{p}=100\;\mu m$.

For convenience, l=50 mm, as it reduces the number of dimensions of the problem and makes the results easily readable. This leaves $h_{e}$, $s_{11}^{e}$ and *w* as parameters to be fine-tuned to optimize the PEA’s morphology and mechanical properties for the SSA’s implementation. The following is the summarized analysis of Figure 4’s subfigures:In Figure 4a, one notices the existence of a global maximum for $f_{1}\left(h_{e},s_{11}^{e}\right)$, that is there is a material $s_{11}^{e*}$ with thickness $h_{e}^{*}$ that guarantees the maximal cantilever tip displacement δ for a given input voltage $v_{c}$. The function $f_{1_{opt}}\left(h_{e},s_{11}^{e}\right)$ describes the optimal combination of $h_{e}$ and $s_{11}^{e}$ to take full advantage of the piezoelectric actuation. The piezoelectric actuation is independent of the cantilever width *w* (see Equation (8)). Therefore, *w* can independently be determined through Equation (11) to guarantee the minimal blocking force.For visualization purposes of $\frac{Q_{\delta }}{v_{c}}\left(w,h_{e},s_{11}^{e}\right)$, one opted for $h_{e}$, $w\in R^{*}$ and four materials as the elastic layer to be used: Cs ($s_{11}^{e}=5.8824\times 10^{-10}$ m${}^{2}$/N), Ag ($s_{11}^{e}=1.2048\times 10^{-11}$ m${}^{2}$/N), Ta ($s_{11}^{e}=5.3763\times 10^{-12}$ m${}^{2}$/N), and Cr ($s_{11}^{e}=3.4602\times 10^{-12}$ m${}^{2}$/N). When analyzing Figure 4b, it is worth noting that $\frac{Q_{\delta }}{\delta_{pre}}\left(w,h_{e},s_{11}^{e}\right)$ is monotonically increasing with respect to *w* and $h_{e}$. This indicates that wider electrodes and thicker elastic layers allow collecting more strain-induced charges. The thickness of the elastic layer is directly linked to the position of the piezoelectric bending neutral plane $h_{\gamma }$ (see Figure 5). The thicker the elastic layer, the more the average stress distribution in the piezoelectric layer becomes positive, that is ${\bar{T}}_{1_{pre}}\left(z_{p}\right)⩾0$. It is known from [34,35] that this positive stress distribution should result in larger stress-induced charges collected through the PEA’s electrodes.Whereas it was not possible to express the blocking force constraint for $f_{1}\left(h_{e},s_{11}^{e}\right)$, it is possible to do so with $\frac{Q_{\delta }}{v_{c}}\left(h_{e},s_{11}^{e},w\right)$. Let us choose Ag as the elastic layer of the cantilever for illustration. Ag is not the material that gives the best sensor sensitivity (see Figure 4b). Nonetheless, Ag has the best electric conductivity among these materials. This allows for the application of $v_{c}$ directly on the elastic layer. This eliminates the need for additional electrodes. As a result, the overall stiffness of the structure and total thickness are not affected. $f_{2}\left(w,h_{e},s_{11}^{e}\right)$ represents the portion of $\frac{Q_{\delta }}{\delta_{pre}}\left(w,h_{e},s_{11}^{e}\right)$ that satisfies the blocking force constraint. The function $f_{2_{opt}}\left(w,h_{e},s_{11}^{e}\right)$ describes the optimal combination of $h_{e}$ and *w* for Ag, as the elastic layer, which offers the best sensitivity to strain-induced charges and satisfies the blocking force geometric constraint.Figure 4d depicts the behavior of the multi-objective function (see Equation (12)), that is the simultaneous sensing and actuation objective where $W_{i}{|}_{i=1,2}=1$. One can choose $W_{i}$ so that either the actuation or the sensor sensitivity is overriding the other. The Pareto front in Figure 6 reveals the trade-off curve between the sensor and actuator objectives, $f_{1}$ and $f_{2}$, respectively.

## 4. Simulation with COMSOL Multiphysics

We realized a stationary study with piezoelectric devices as the physics to be used and built a clamped unimorph piezoelectric bender. A constant input voltage of $v_{c}=60$ V was applied to the PEA and the value of the tip deflection, and colormap of the stress distribution in the x−direction, $T_{1_{pre}}\left(x,y,z\right)$, thereof, is displayed in Figure 7. For the readability of the results and comparison purposes with the analytical model, we chose to keep only *w*, $h_{e}$ as variable parameters. The piezoelectric material to be used for this optimization was PZT-5H with a fixed layer thickness of $h_{p}=100\;\mu$m. Ag was chosen as the elastic layer, and the cantilever length was fixed at l=50 mm. The piezoelectric material in the actuator, PZT-5H, and elastic layer material, Ag, were already defined in the material library of COMSOL Multiphysics.

A parametric sweep study through a defined range was performed in COMSOL Multiphysics to determine the influence of *w* and $h_{e}$ in the actuation (PEA’s tip displacement δ) and the sensor sensitivity to strain due to piezoelectric actuation $\left(\frac{Q_{\delta }}{\delta_{pre}}\right)$. Figure 8 depicts simulation results performed in COMSOL Multiphysics considering the same unimorph piezoelectric cantilever as in the previous section. As for the analytical model (see Figure 4a,b), the PEA’s actuation presented a global maximum and was independent of the cantilever width *w*. The strain-induced charges were monotonically increasing with respect to *w* and $h_{e}$. However, it must be pointed out that the analytical model of the unimorph piezoelectric cantilever introduced by [36] and adopted by [34,35,38] treats in-plane problems. This model only considers stress in the *x*-direction, $T_{1}\left(x,y,z\right)$. According to the linear theory of piezoelectricity [39], the electric displacement, $D_{3}$, should include stress in the *y* and *z*-direction, $T_{2}\left(x,y,z\right)$ and $T_{3}\left(x,y,z\right)$, respectively. It is worth mentioning that $T_{i}{|}_{i=1,2,3}\left(x,y,z\right)$ are not applied external stresses, but rather, stresses caused by the piezoelectric converse effect under the application of $v_{c}$. The strain-induced charges should thus be given by: (14)$$Q_{\delta }\;\left(T_{1},\;T_{2},\;T_{3}\right)=\frac{1}{h_{p}}\int_{h_{p_{-}}}^{h_{p_{+}}}\overset{l}{\underset{0}{\int }}\overset{w}{\underset{0}{\int }}\left\{d_{31}\left[T_{1}\left(x,y,z\right)+T_{2}\left(x,y,z\right)\right]+d_{33}T_{3}\left(x,y,z\right)\right\}\mathrm{dxdydz}$$

Figure 8b depicts both the strain-induced charges when considering only $T_{1}\left(x,y,z\right)$ and the one resulting from Equation (14). A finite element analysis considering the lateral deformation, rather than the 2D boundary conditions of the analytical model, treats the coupling problem more rigorously and furnishes more realistic results. It is worth noting that both $Q_{\delta }\left(T_{1}\right)$ and $Q_{\delta }\left(T_{1},T_{2},T_{3}\right)$ are monotonically increasing with respect to *w* and $h_{e}$ and only differ in magnitude.

Whereas choosing the PEA’s width *w* will only impact the sensor sensitivity to strain, there is a trade-off between the PEA’s actuation and sensor sensitivity when it comes to the elastic layer thickness $h_{e}$ (see Figure 6a). Therefore, it may be necessary to choose to favor either the actuation or the sensitivity of the sensor, according to the needs of the application.

## 5. Experimental Validation

Nine unimorph piezoelectric cantilevers were used for the validation of the numerical and analytical analysis. They were made of an upper piezoelectric layer with fixed thickness $h_{p}=100\;\mu$m and a lower nickel layer with a thickness $h_{e}=\left\{50,\;100,\;200\right\}$ (μm). For each $h_{e}$, a width w={1.5,2.5,3.5} (mm) of the cantilever is associated. The nickel foils (NI000450, NI000480, NI000550) were purchased at Goodfellow, then diced into the aforementioned dimensions. The piezoelectric layer (PZT-5H) was purchased at PIceramic. A Step input voltage $v_{c}=100$ V was applied to the PEA. During the application of $v_{c}$, the PEA’s tip displacement δ(t) was measured with the displacement optical sensor LC-2420, and the current going through the PEA ${\dot{Q}}_{pea}\left(t\right)$ was measured with the current probe HIOKI CT6700. The algorithm presented in [13] was implemented on the real-time hardware dSPACE (DS1103) for the extraction of the strain-induced charges $Q_{\delta }$ from ${\dot{Q}}_{pea}$.

The experimental setup used for this validation is illustrated in Figure 9a. Figure 9c,d present the fitted surfaces of the measured PEA’s tip displacement $\bar{\delta }$ and strain-induced charges $\bar{Q_{\delta }}$ resulting from the application of $v_{c}$. $\bar{\delta }$ and $\bar{Q_{\delta }}$ are displayed as dots in these figures. They respectively represent the average of δ(t) and $Q_{\delta }\left(t\right)$ under 50 seconds from the application of the step input voltage $v_{c}$.

Results displayed in Figure 9 corroborate the numerical and analytical conclusions, that is there is a trade-off between the PEA’s actuation and sensor sensitivity in the choice of the elastic layer thickness $h_{e}$. Increasing the PEA’s sensor sensitivity will be at the cost of its actuation and vice versa. In this case, the objective function (see Equation (12)) is said to be conflicting, and there exists a (possibly infinite) number of Pareto optimal solutions (see Figure 6). Without a piece of additional subjective preference information on either the PEA’s actuation or sensor sensitivity, all Pareto optimal solutions are considered equally good.

## 6. Conclusions

This paper presented an analytical optimization approach of a unimorph piezoelectric bender for charge-based SSA. For a chosen piezoelectric material, a thickness was imposed by its depolarizing field. The suggested approach helped fine-tune the choice of the thickness and material of the elastic layer, as well as the width of the entire piezoelectric bender. These choices have led to a set of solutions that optimizes the piezoelectric actuation and its sensitivity to strain-induced charges (due to mechanical load and/or piezoelectric actuation). The final design resulted from the optimization of a multi-objective function, which aimed at a better stress distribution in the piezoelectric layer for better strain-induced charge sensitivity and the structure’s stiffness for maximum deflection as a response to input voltages. A finite element analysis (FEA) implemented in commercial software was used to corroborate the analytical approach. Finally, experiments were carried out to validate the FEA results.

The analytical, numerical, and experimental results have led to the same conclusion: there is a trade-off between the PEA’s actuation and sensor sensitivity when implementing a charge-based SSA in a piezoelectric cantilevered actuator. Therefore, depending on the application, a piece of additional preference information on either the PEA’s actuation or sensor sensitivity must be provided.

## Figures and Tables

**Figure 1 sensors-19-02582-f001:**
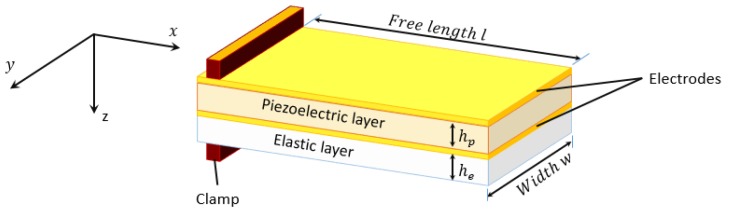
Layer sequence of a unimorph.

**Figure 2 sensors-19-02582-f002:**
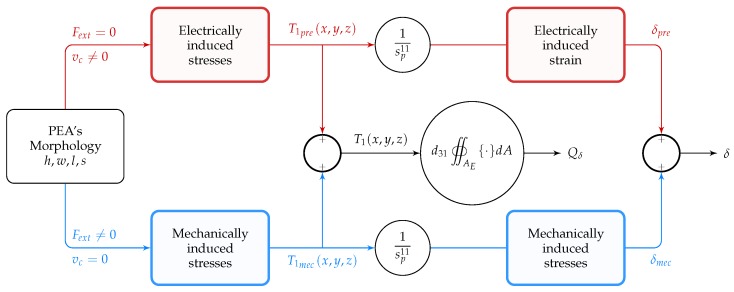
Schematic of the superposition of bending under external loading (—) and piezoelectric actuation (—). PEA, Piezoelectric Actuator.

**Figure 3 sensors-19-02582-f003:**
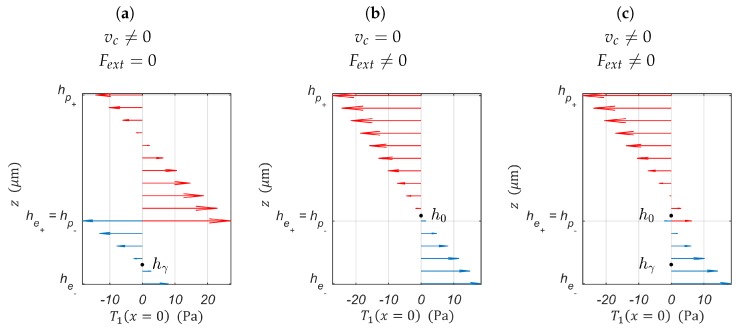
Stress distribution of a cantilevered piezoelectric unimorph ($h_{p}=100$
μm and $h_{e}=50$
μm): → piezoelectric layer’s stress distribution and → elastic layer’s stress distribution. The piezoelectric material used for this stress distribution is the PSI-5H4E ($d_{31}=-320\times 10^{-12}$ C/N, $s_{11}^{p}=1.6129\times 10^{-11}$ m${}^{2}$/N), and the elastic layer is Ni ($s_{e}^{11}=4.67\times 10^{-12}$ m${}^{2}$/N).

**Figure 4 sensors-19-02582-f004:**
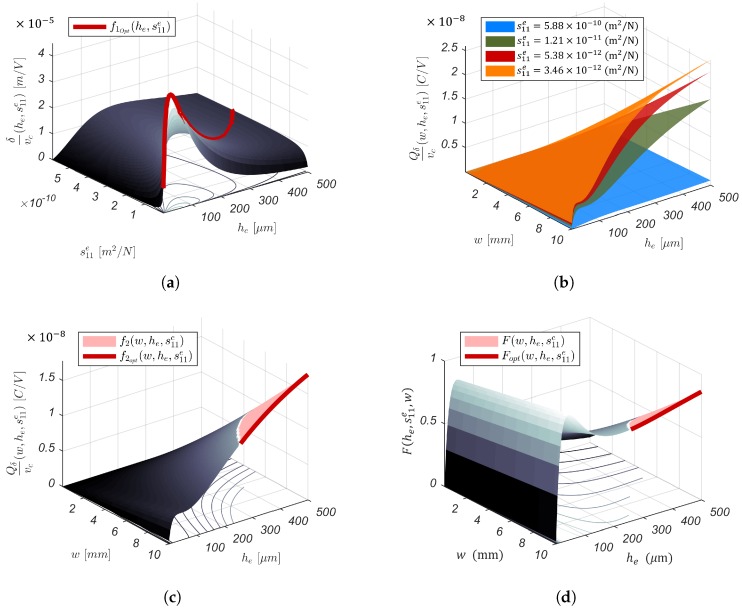
Geometric parameters and material properties’ dependence. (**a**) Piezoelectric actuation. (**b**) Piezoelectric sensor’s sensitivity to strain induced by piezoelectric actuation for Cs, Ag, Ta, as Cr as the elastic layer of the bender. (**c**) Piezoelectric sensor’s sensitivity to strain induced by piezoelectric actuation for Ag as the elastic layer of the bender. (**d**) Simultaneous sensing and actuation: ”Self-Sensing”.

**Figure 5 sensors-19-02582-f005:**
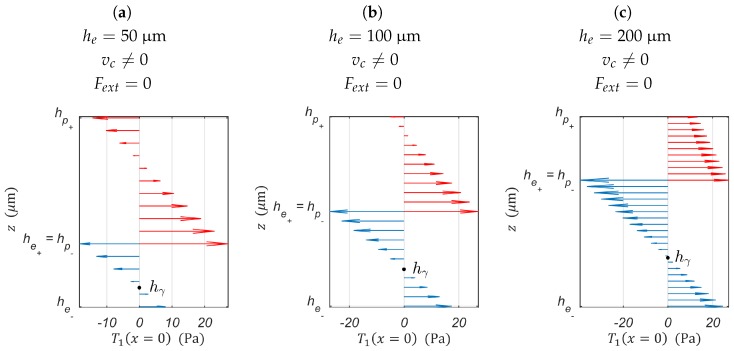
Piezoelectric actuation neutral axis evolution due to elastic layer thickness change. The piezoelectric layer is made of PSI-5H4E, and the elastic layer is Ag. The piezoelectric layer thickness is fixed at $h_{p}=100\;\mu$m.

**Figure 6 sensors-19-02582-f006:**
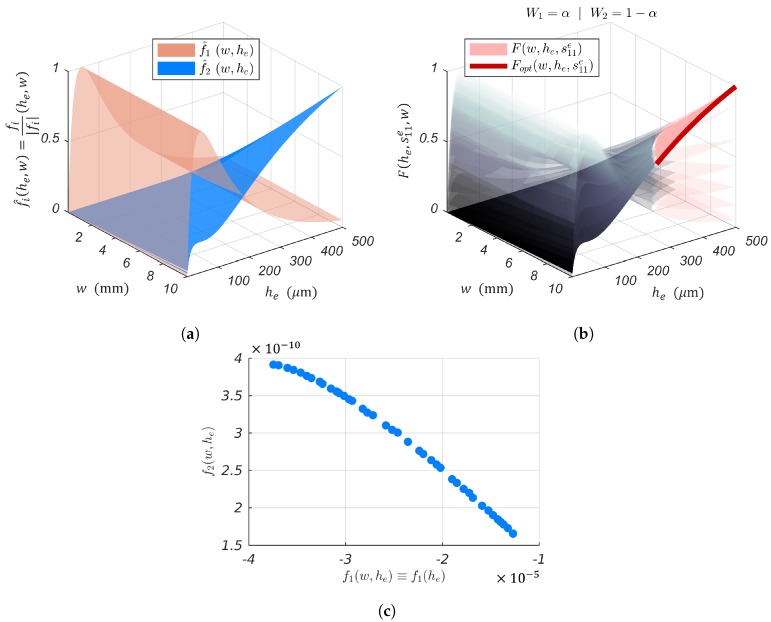
Trade-off between the actuator and sensor objectives, $f_{1}$ and $f_{2}$, respectively. The piezoelectric actuation $f_{1}$ is independent of the cantilever width. (**a**) Overlapping of the PEA’s actuation and sensing behaviors. (**b**) Evolution of the SSA’s geometrical optimal configuration as a function of the weighting factor $W_{i}$. $F(w,h_{e},s_{11}^{e})$ describes combinations of $h_{e}$ and *w* for Ag as the elastic layer that satisfy the blocking force geometric constraint. $F_{opt}\left(w,h_{e},s_{11}^{e}\right)$ represent the optimality of $F(w,h_{e},s_{11}^{e})$. (**c**) Pareto front.

**Figure 7 sensors-19-02582-f007:**
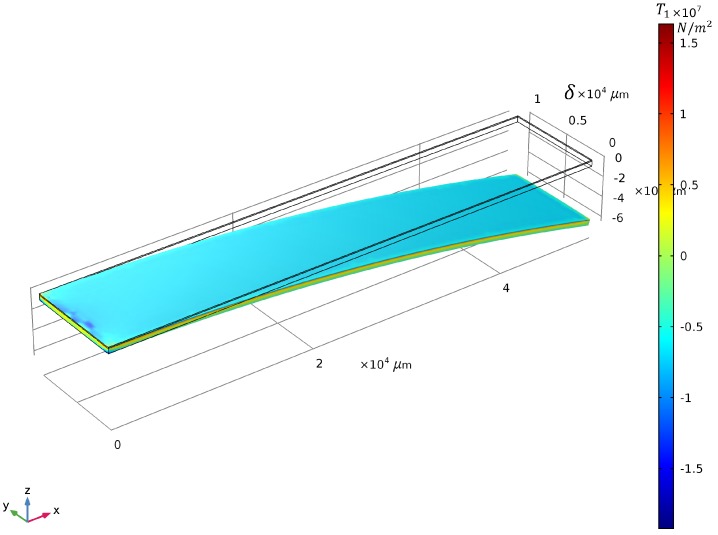
Colormap of the stress distribution in the *x*-direction
$T_{1_{pre}}\left(x,y,z\right)$ (N/m${}^{2}$) at $v_{c}=60$ V and illustration of the bender tip deflection (for this figure, $h_{e}=200\;\mu$m and w=10 mm).

**Figure 8 sensors-19-02582-f008:**
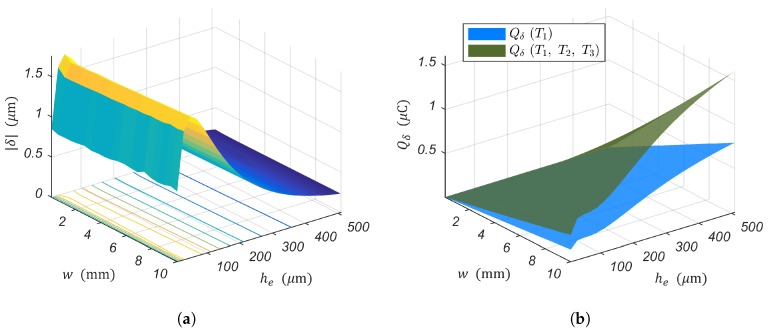
Numerical evaluation of the influence of the varying PEA’s width *w* and elastic layer thickness $h_{e}$. (**a**) The piezoelectric actuation. (**b**) The piezoelectric sensor’s sensitivity to strain induced by piezoelectric actuation.

**Figure 9 sensors-19-02582-f009:**
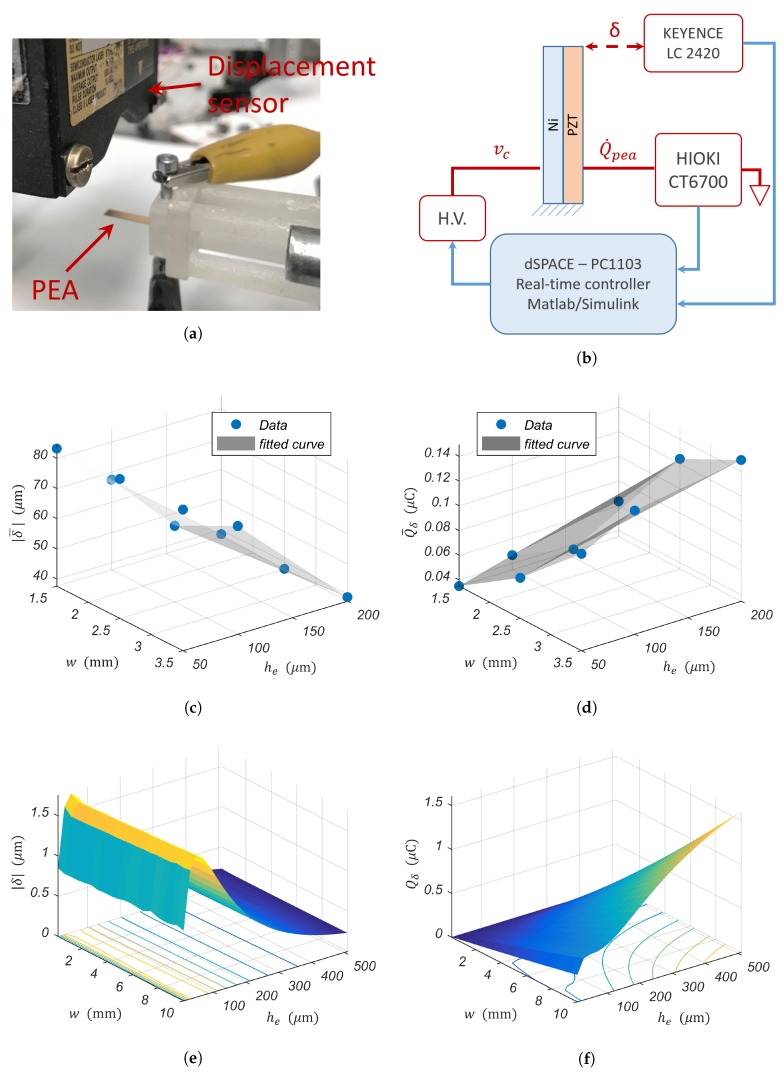
Influence of varying PEA’s width *w* and elastic layer thickness $h_{e}$: measured, numerical, and analytical results. (**a**) Photo of the experimental setup. (**b**) Schematic block diagram of the experiment. (**c**) Measured PEA’s tip displacement due to the application of a step input voltage: PEA’s actuation. (**d**) Measured strain-induced charges due to the application of a step input voltage: PEA’s sensor sensitivity. (**e**) Numerical evaluation of the piezoelectric actuation. (**f**) Numerical evaluation of the PEA’s sensor sensitivity. (**g**) Analytical evaluation of the piezoelectric actuation with the application of the geometric constraint (Equation (11)) to guarantee the minimal blocking force, $F_{bl}>100\;\mu$m. (**h**) Analytical evaluation of the PEA’s sensor sensitivity.

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
