# Peer review of "Optimal Design of Piezoelectric Cantilevered Actuators for Charge-Based Self-Sensing Applications"

_sensors, 2019, doi:10.3390/s19112582_

Reviewer 1 Report

The manuscript reports an analytical optimization approach of an unimorph piezoelectric bender for charge-based SSA. The research is systematic and complete. The design optimization is carried out using a multi-objective function which aims for better strain-induced charges sensitivity and maximum deflection as a response to input voltages. The proposed analytical approach is verified by the finite element analysis, which is further validated by experimental results. The text of the manuscript is well-written, and figures are well-designed. Some minor issues need to be addressed before this manuscript becomes acceptable for publication:

1.         It is better to provide the basic principle of SSA at the beginning of section 2. For example, authors state “To improve the SSA performance …” at the beginning of section 2.1, but readers, who are not expert in this field, can be confused since they do not know how to define “SSA performance”. Authors directly go to optimization of position of h0 and hr without providing some basics.

2.         The basic principles of two methods, which are based on charge equations and displacement equations, should be described briefly.

3.         In figs 3, 4 and 5, some captions of sub figures are truncated.

4.         Is it possible to include the numerical and analytical results in fig 9 so that it becomes more straightforward to compare these results?

Author Response

Dear Reviewer,

Thank you for the opportunity to revise our manuscript submitted to Sensors entitled: ‘Optimal design of piezoelectric cantilevered actuators for charge-based Self-Sensing Applications .’ The suggestions that you offered have been immensely helpful.

We have included your comments immediately after this letter and responded to them individually, indicating exactly how we addressed each concern or problem and describing the changes we have made.

Here is a point-by-point response to the reviewer’ comments and concerns.

1. An introduction on Self-Sensing Actuation (SSA) is provided at the beginning of section 2. This introduction furnishes the background that one needs to know while using piezoelectric materials as transducers and what characterizes SSA’s performances. The fundaments introduced herein are pivotal for the understanding of reasonings and arguments presented in subsection 2.1 and 2.2.

2.From this insightful comment of the reviewer, we introduced the basic principles of charge and displacement equations at the end of section 2. This will prepare the reader for the detailed explanation provided for the case addressed in this paper: unimorph piezoelectric cantilever.

3. Modifications have been made so that the captions in figures 3, 4 and 5 are no longer truncated.

4.We included the numerical and analytical results in figure 9. As the reviewer has imagined it, this change made it easier to compare the measured, analytical and numerical results. The reader will not need to navigate back and forth between pages for the comparison.

We hope the revised manuscript will better suit to Sensors but are happy to consider further revisions, and we thank you for your continued interest in our research.

Sincerly,

Reviewer 2 Report

The paper reports a piezoelectric cantilever as actuator and sensor, involving modelling and experimental validation. The model and experiments are professionally carried out and well presented. However, I fail to see the novelty of this paper, or any original contribution to the literature. Piezoelectric transducers, especially in cantilever form, is well known in the literature, as generators, sensors and actuators. For example, self sensing piezoelectric accelerometers have been commercially used for many decades now, and it is a very well known, well characterised technology.

Furthermore, FEA simulation in COMSOL didn't seem to use the piezoelectric device physics, and only the solid mechanics module. COMSOL simulation involving both the electrical and mechanical multiphysics of piezoelectric actuators have also already been reported. So a basic mechanical FEA has little value, even if it is using the PZT material property. I would encourage the authors to highlight their unique focus and how this paper aims to address any gaps in literature. Below are a few minor points:

- caption in fig9b cut off, fig4a as well, 

- fig 7 talks about stress, but the units are in displacement? it would be useful to include comsol title in the plot, and unit for the colour bar, showing what simulation results we are actually looking at? stress or displacement? If stress, which stress? von mises stress?

- derivative operator d in equations shouldn't be italic. e.g, the d in dx dy dz.. it is an operator, not a parameter.

- More details on the piezoelectric cantilevers used for the experiments would be useful. Their materials (piezo, substrate, electrodes?), how they were fabricated, the cross sectional stack?

Author Response

We sincerely thank the reviewer for constructive criticisms and valuable comments, which were of great help in revising the manuscript. Accordingly, the revised manuscript has been

systematically improved with new information and additional interpretations.

Here is a point-by-point response to the reviewer’ comments and concerns.

0.1.We agree with the reviewer that Self-Sensing Actuation (SSA) has been around for quite a time now. Different approaches have been implemented for simultaneous actuation and sensing in the literature.The way the text was presented may appeared more ambiguous than intended, sowe have adjusted the text to make our contribution clearer.Section1have been divided into subsections that specifies the state of the art in piezoelectric actuation, sensor and simultaneous actuation/sensor optimization so that our contribution can easily be pointed out.

0.2.This is a valid point. Indeed, we used the piezoelectric device physics. Beside the piezoelectric device physics, the other way to establish an electromechanical coupling would be writing the transduction relation by hand. The result would still be the same if the coupling coefficients are the same. Since there are at least 2 ways of expressing a transduction relation between two domains (in this case electrical and mechanical) we did not consider it necessary to impose a way of doing things but rather the objective to be achieved. The numerical results that we presented were intended to be the stepping stone to the experiments. The analytical equations encountered in the literature are limited to 2D boundary conditions. In order to treat the coupling problem more rigorously and furnish more realistic results a 3D implementation of the problem using a numerical study (to avoid the deduction of long and complex analytical equations) was necessary before performing experiments.

1.Modifications have been made so that the captions in figures are no longer truncated.

2.Additional information can be observedin the figure 7to differentiate the displacement from the stress. As mentioned in the caption of figure 7, it is the stress in the x-direction. The von Mises stress is a criterion for yielding that combined stress in different directions to predict failure. Failure study is an interesting aspect but we feel that it falls outside the scope of this study. The equation 16 depicts the 3 stresses that are used for the calculation of the strain-induced charges.We hope that clarifies the doubt. If not, please, we welcome any suggestion to make the paper clearer.

3.The operator are no longer in italic.

4.We provide the nomenclatures and manufacturers of the Piezoelectric material and elastic layer (Nickel) that are usedfor the experiments.With these information the reader has access to all the necessary information about how they were fabricated.

We hope the revised manuscript will better suit to Sensors but are happy to consider further revisions, and we thank you for your continued interest in our research.

Sincerly,

p { margin-bottom: 0.25cm; line-height: 120%; }

Round  2

Reviewer 2 Report

I am not fully convinced by the authors' response and updates. All the information the authors added to sections 1 and 2 to strengthen the original contribution claims for this paper, is still not new. In fact, many of these optimisation techniques are well known and used for decades in the wider discipline, and it has been applied specifically to piezoelectric transducers too. The fundamental point of the paper has not been fully addressed.

Also the authors said they used piezoelectric device and coupling in FEA, but this is not clearly shown in the paper. Using piezoelectric material is not the same as setting up a multiphysics simulation. There should be evidence showing the inclusion of pieoelectric and ES physics in the model. I would advise the authors to include more method details in section 4, including more description and images. Plots of electrical responses and inputs into the multiphysics model would be useful too.

Author Response

We would like to thank the reviewer for careful and thorough reading of this manuscript and for the thoughtful comments and constructive suggestions, which help to improve the quality of this manuscript. Our response follows.

To optimize the design of piezoelectric devices for Self-sensing Actuation (SSA), it is important to account for the phenonmena involved, their dependences and/or existing conflicts.
The additional information that was introduced in section 1 (1.1 - 1.3) after the first round of the reviewer comments was intended to differentiate simultaneous actuation and sensing optimization encountered in the literature for which actuation and sensing do not share the same electrodes from SSA for which the same electrodes are used for both sensing and actuation.
In that regard, to the best of our knowledge, there is no existing work. We presented an analytical approach based on a multi-objective optimization and backed that  up the actuation and sensing behavior with experiment resluts.

The reviewer is right in saying that all the optimization techniques used in this paper have been already implemented in the literature for piezoelectric elements (actuator and sensor).
The originality of this paper is in showing that there is no unique optimal solution for SSA's implemenation in the case of unimorph piezoelectric bender, but rather a (possibly infinite) number of Pareto optimal solutions.

This paper suggests that an additional criteria presented as a weights (ponderation) of the two objective functions (see Eq. 14) is required for this optimization. However, it is clear that if the reviewer, who is an expert in the field, says that the fundamental point of the paper has not been fully address it is because of the presentation of results.
As an attempt to correct that mistake we introduced a third subfigure in figure 6. This subfigure would facilitate the visualization of the multiobjective optimization (see Eq. 14) and help understand the importance of the ponderation $W_{i}$ in the determination of the optimal geometrical characteristics (in this case width and thickness of the elastic layer) of the unimoprh piezoelectric bender for SSA's implementation.

We are deeply sorry for not have understood the first attemp of the reviewer to alert us that our numerical study was vague. A phrase is added at the begining of section 4 to indicate that we realized a stationary study with piezoelectric devices as physics to be used  and build a clamped unimorph piezoelectric bender.   
A constant input voltage of $v_{c} =60~V$ is applied to the PEA and the value of the tip deflexion and colormap of the stress distribution in the $x-direction$ thereof are displayed in figure 7. Unfortunately, because it is a stationary study and the piezoelectric material behaves as a capacitor (lower frequencies) the measured current will be null and the input voltage a static value. Measuring charges will include the dielectric and stress-induced charges (see Eq. 5). For this particular case, we eliminated the dielectric charges using directly Eq. 5 for 2D and 3d considerations. For the experiment we used a mean of 50 sec after the transient part for comparison with the numerical and analytical results.